# Understanding the friction of atomically thin layered materials

David Andersson [1,2] & Astrid S. de Wijn [1,2]*

Friction is a ubiquitous phenomenon that greatly affects our everyday lives and is responsible for large amounts of energy loss in industrialised societies. Layered materials such as graphene have interesting frictional properties and are often used as (additives to) lubricants to reduce friction and protect against wear. Experimental Atomic Force Microscopy studies and detailed simulations have shown a number of intriguing effects such as frictional strengthening and dependence of friction on the number of layers covering a surface. Here, we propose a simple, fundamental, model for friction on thin sheets. We use our model to explain a variety of seemingly contradictory experimental as well as numerical results. This model can serve as a basis for understanding friction on thin sheets, and opens up new possibilities for ultimately controlling their friction and wear protection.

---

[1] Chemical Physics, Department of Physics, Stockholm University, AlbaNova University Center, SE-106 91 Stockholm, Sweden. [2] Department of Mechanical and Industrial Engineering, Faculty of Engineering, Norwegian University of Science and Technology (NTNU), 7491 Trondheim, Norway. *email: astrid.dewijn@ntnu.no

Considering that approximately 23% of the world's energy consumption[1] is due to friction, there is an urgent need for better low-friction technologies and greater understanding of friction and lubrication (tribology) both at macroscopic and microscopic scales. Layered materials are of great interest in this context, because they typically have low friction. They are often used as (additives to) lubricants or coatings. Moreover, thin sheets of graphene have the potential to be used as wear protection[2]. Development and implementation of new technologies, however, is hampered by our lack of basic fundamental understanding, especially at the nanoscale. Nevertheless, in recent decades, major progress has been made due to the development of the atomic force microscope (AFM), as well as increases in computing power that now allow massive atomistic simulations.

AFM experiments on atomically thin sheets, comprised of one or more layers of graphene or other layered materials[3,4], have shown that the friction depends on the number of layers in a surprising way: it is highest for single-layer sheets and decreases with increasing number of layers. In some experiments, an initial strengthening effect has also been observed, where the friction increases slowly at the onset of motion and then reaches a plateau[4]. This effect is also stronger for sheets consisting of fewer layers and appears to be related to the higher friction. These effects appear to be very general and related to the layered structure, as they have been detected in a variety of materials.

A number of mechanisms have been proposed for this peculiar behaviour and have been investigated experimentally[5–7] as well as in detailed molecular dynamics (MD) simulations[8]. These investigations have led to some controversy and discussion[9], because different AFM experiments and MD simulations from different authors have produced different suggestions and conclusions about what kind of mechanisms play a role here.

Initially, it was suggested that the experimental results could be explained by some kind of out-of-plane bending such as wrinkling and puckering[4]. Thicker sheets are more rigid, and thus any effects of bending of the sheet should become smaller when the number of layers increases. The out-of-plane idea was both confirmed in some MD simulations[8] and disproved in others[10]. In the latter work, it was suggested that a kind of evolving quality of the contact was the origin of the strengthening. Other suggestions have resolved around, e.g. delamination[11]. To elucidate the effect of the substrate, friction experiments have been performed on different substrates[5] and suspended graphene sheets (i.e. with no substrate)[12]. In some cases, it was found that the layer-number dependence disappeared for strongly bound sheets[5], while in some simulations[4] the opposite was found. Suspended systems, moreover, produce unexpected results with friction decreasing at higher loads[12,13], which has been suggested to be related to a reduction in out-of-plane bending.

In this work, we propose a new model for friction on atomically thin sheets that we use to explain all these experimental and simulation results. A sketch of the model system we study is shown in Fig. 1. The model is based on the Prandtl–Tomlinson (PT) model[14,15], with the addition of one extra degree of freedom, which can represent, e.g. bending or some in-plane degree of freedom such as delamination. We construct this model by systematically expanding the contribution to the potential energy landscape due to the distortion. Because of its simplicity, this model allows us to isolate and understand the dynamics of strengthening and layer-number dependence of friction. We also use it to investigate the influence of various potential mechanisms as well as the role of the substrate and system parameters.

With the help of our model, we establish a close connection between the sheet–substrate geometry and the resulting friction response. We show that our model can explain and unify the

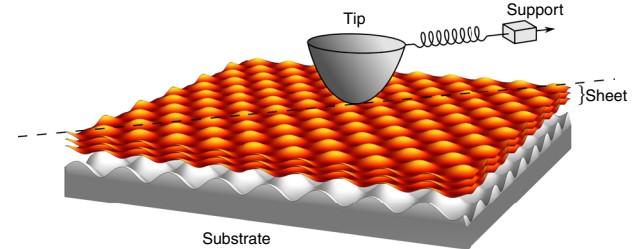

**Fig. 1 A sketch of the model system as studied in this work.** A tip is dragged via a spring over a sheet comprised of a number of atomically thin layers that lie on a substrate but do not slide on it.

various experimental and atomistic numerical results and that the seeming contradictions that have been found are simply the result of different degrees of freedom giving rise to similar dynamics.

## Results

**The model.** The original PT model[14,15] has been very successful in providing understanding of atomic friction in a wide range of systems[16]. Notably, it has been used widely to model AFM experiments. It consists of a point particle (the tip) moving in a periodic potential (the substrate). The tip is pulled via a spring (the tip/cantilever elastic deformation) by a support moving at constant velocity (the stage). The dissipation into phonon modes in the substrate is modelled by a viscous damping term on the tip. The PT model captures an important characteristic of friction on all length scales: stick–slip motion. For sufficiently weak springs and sufficiently slow sliding speeds, the particle will periodically be stuck in the minima of the potential and slip over the potential barriers when enough force has built up in the spring. The friction is the average lateral force needed to keep the support moving, which is equal to the force in the spring. As useful as the PT model has been for understanding nanoscale friction, there are many things that it does not capture. One such set of phenomena relates to atomically thin layered materials, which are the scope of this paper.

The model we propose in this paper is similar to the ordinary PT model, but with the addition of one extra degree of freedom $q$ that describes the internal dynamics of a sheet, composed of one or more atomically thin layers. This degree of freedom may represent bending, shearing, or other deformation, including the highly nontrivial "quality of the contact" reported in ref. [10]. Since this distortion is given by a displacement of the atoms, we take it to have the dimension of length. The coupling between $q$ and the tip position $x$ is governed by the tip–sheet ($V_{\text{tip−sheet}}(x, q)$) and tip–substrate ($V_{\text{tip−substrate}}(x, q)$) interactions. Moreover, when $q$ is nonzero, the sheet has a distortion energy denoted by $V_{\text{sheet}}(q)$. The total potential energy of the system then becomes

$$U(x, q, t) = \frac{1}{2} k(x - vt)^2 + V_{\text{sheet}}(q) \\ + V_{\text{tip−sheet}}(x, q) + V_{\text{tip−substrate}}(x, q). \quad (1)$$

The first term on the right-hand side represents the flexibility of the AFM tip and cantilever and consists of a spring with spring constant $k$ between the tip and support moving at constant velocity $v$.

We perform a systematic coarse graining of the sheet by expanding the energy in $q$. The distortion energy of the sheet $V_{\text{sheet}}(q)$ should be at a minimum for $q = 0$. We keep the quadratic leading order and fourth-power next-to-leading order in the expansion, so that

$$V_{\text{sheet}}(q) = \nu_2 q^2 + \nu_4 q^4. \quad (2)$$

As we show below, it is important to keep the next-to-leading order, as it is crucial for limiting the $q$ dynamics.

The tip slides over the periodic sheet. As the sheet deforms, the energy barriers that the tip must overcome to slide over the sheet change. Hence, the corrugation depends on $q$. Since the corrugation should be at a minimum for an undistorted sheet, this dependence is to leading order quadratic. These interactions are given by

$$V_{\text{tip}-\text{sheet}}(x,q) = (V_1 + \kappa_1 q^2)\left(1 - \cos\left[\frac{2\pi}{a}(x - q)\right]\right), \quad (3)$$

where $V_1$ is the corrugation for an undistorted sheet, $\kappa_1$ accounts for the change due to the distortion, and $a$ is the sheet lattice parameter. In principle, the distortion may lead to a phase shift in the periodicity of the tip on the sheet. Without loss of generality, we have set the coefficient of this to 1.

The tip–substrate interaction is weaker than the tip–sheet interaction, but an atomically thin sheet cannot mask the substrate completely. Similar to the tip–sheet interaction, if the substrate is periodic, the tip–substrate interaction is periodic in $x$ as well. Moreover, as the sheet is distorted, the transmission of the substrate corrugation through the sheet changes, and the barriers it must overcome change accordingly. This interaction is given by

$$V_{\text{tip}-\text{substrate}}(x,q) = (V_2 + \kappa_2 q^2)\left(1 - \cos\left[\frac{2\pi}{b}x\right]\right), \quad (4)$$

with $V_2$ and $\kappa_2$ playing the same roles as $V_1$ and $\kappa_1$ and $b$ the lattice parameter of the substrate. However, since the substrate is fixed in place, there can be no phase shift in the tip–substrate interaction.

More details on the implementation of the model and chosen parameters are given in the "Methods" section. We note, however, that the stick–slip behaviour in the PT model, and along with it also the strengthening in our model, is quite robust against changes in parameters.

**Simulation results**. A typical force trace is shown in Fig. 2a. In this case, the lattice periods of the substrate and sheet were the same. The lateral force is plotted as a function of support displacement (time), along with the sheet distortion. The system exhibits stick–slip friction, as expected for these parameter values based on the PT model. In the initial stages, however, the slips do not all happen at the same lateral force. There is a buildup (strengthening) of the friction over several stick–slip periods until

a steady state is reached, similar to what is found in experiments[4] and detailed simulations[10]. Meanwhile, as can be seen from Fig. 2b, the sheet distortion changes only very little during sticks, but at slips during the strengthening there is an abrupt shift to a larger sheet distortion. Figure 2c shows the trajectory in the $xq$-space, superimposed on a heat plot of the potential energy contribution from the tip–substrate and tip–sheet interactions. The sticks correspond to spending time near the energy minima in this two-dimensional energy landscape.

In this model, we can include the thickness of the sheet by noting that thicker sheets will have a higher stiffness (bending or otherwise), which is manifested in our model parameter $\nu_4$. We do not expect the $\nu_2$ coefficient to depend strongly on the number of layers, since under realistic conditions, it would be dominated by the adhesion between the sheet and substrate. If we assume that each layer is distorted in roughly the same way, the energy penalty should grow roughly linearly with the number of layers $n$ and thus $\nu_4$ is proportional to $n$. There are other effects, such as changes in $V_2$ and $\kappa_2$, but as we will discuss in more detail later, we find that these do not strongly affect the steady-state friction. Figure 3 shows the steady-state force as a function of the parameter $\nu_4$ for both the numerical simulations and the analytical expression in Eq. (7) (explained below) for an incommensurate sheet and substrate with maximum incommensurability $a/b = \frac{1}{2} + \frac{1}{2}\sqrt{5}$. This figure thus shows how our model predicts that the friction should depend on the number of layers. While it is possible for $\nu_4$ to contain second-order corrections in $n$, this would not qualitatively change the picture. This dependence corresponds remarkably well to the reported decrease in the friction on thicker sheets[3,4,10,13,17], which we include also in the figure for comparison. The experimental results show a decrease in friction of approximately 20% between single and bilayer sheets and that this decrease then rapidly decays with number of sheets. We find the same behaviour very robustly from our model, without any need for tuning the parameters.

Figure 4 is similar to Fig. 2 and shows a force trace, sheet distortion, and $xq$-trajectory but for lattice parameters that are (weakly) incommensurate. This corresponds to the case of atomically thin sheets showing moiré patterns. Owing to the superposition of the two periodic functions with incommensurate periods, which leads to the long-range periodicity of the moiré pattern, the resulting force trace exhibits a long-range periodicity. It

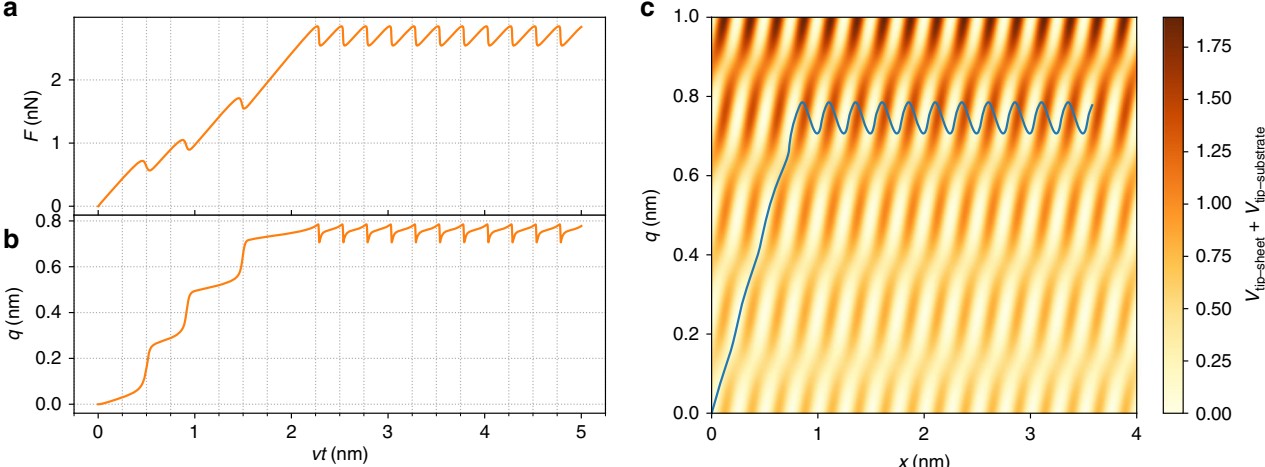

**Fig. 2 Typical behaviour of our model. a** Lateral force as a function of time, **b** sheet distortion as a function of time, and **c** a trajectory in $xq$-space showing the position of the tip and simultaneous distortion of the sheet. The strengthening is similar in behaviour to that found in experiments[4] and detailed simulations[10]. The frictional strengthening is coupled to, and limited by, the sheet distortion. The strengthening behaviour of the system can be understood from the potential landscape.

still displays the same strengthening and steady-state behaviour. Moreover, the trajectory in the $xq$-space also remains very similar, with the sheet distortion approximately constant once the steady state is reached but without the periodicity of the commensurate case. However, the long-range periodicity of the supercell exhibits a significantly different force profile compared to a rigid sheet where $q = 0$ is fixed (also included in the figure), which would show a sinusoidal modulation. Such an effect has indeed been observed in friction on moiré patterns[18].

**Understanding strengthening and layer dependence.** We can understand the strengthening behaviour and other effects described above by considering the trajectory in the $xq$-space.

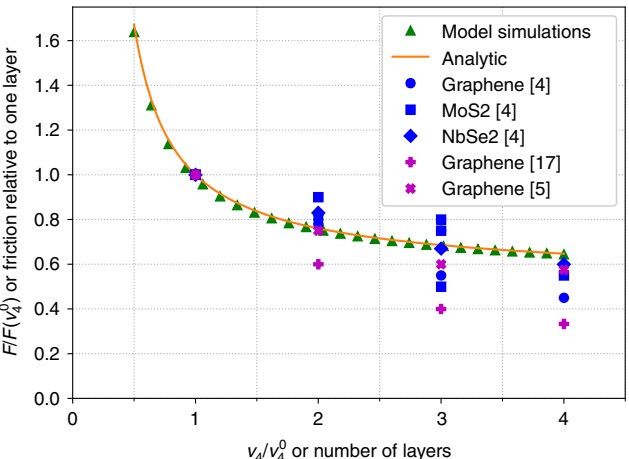

**Fig. 3 The steady-state friction.** The friction obtained numerically from our model with incommensurate lattice parameters (green triangles) along with the analytical estimation in Eq. (7) (orange line) and several literature results from experiments (filled blue circles, squares, and diamonds) and detailed atomistic simulations (purple crosses). $\nu_4$ scales linearly with the number of layers, $n$. Our standard value $\nu_4 = 3.64$ eV nm$^{-4} = \nu_4^0$ corresponds roughly to a single layer. All values are scaled by the friction on a single layer, and hence all points for $\nu_4/\nu_4^0$ fall on top of each other.

We first note that the plain PT model in this regime is quasi-static, i.e. the tip remains near a local energy minimum for long intervals while the support moves slowly away. Once the force in the spring is large enough to pull the tip over the energy barrier, there is a sudden slip to a new, usually adjacent, energy minimum. This picture is independent of the velocity, as long as the velocity is sufficiently low. The addition of the extra degree of freedom in our model makes the energy landscape two dimensional, rather than one dimensional. However, the quasi-static behaviour remains fundamentally the same. We note that this means that the stick–slip and strengthening in our model are not sensitive to the choice of parameters, in the same reason that stick–slip is robust in the one-dimensional PT model. The tip still sticks in the minima in the two-dimensional energy landscape and slips from one minimum to the next. Thus the trajectory in $xq$-space is determined by the structure of the minima, which therefore determine all of the behaviour in this regime.

Figure 5 shows a simple sketch of this $xq$-energy-landscape topology that is helpful for understanding. There are slanted trenches in the potential landscape originating from the tip-sheet interaction term, as can also be seen in Figs. 2 and 4. As the tip is pulled in the $x$ direction, these trenches lead it away from $q = 0$. For sufficiently large sheet distortions, the energy penalty for further distorting the sheet ($V_{\text{sheet}}(q)$) will become very high. Note that the sheet-distortion contribution to the potential energy is not plotted in the figures, in order to enhance the contrast. While moving through the trench, the tip encounters successive minima that it gets stuck in and saddle points that it slips over. This can be seen in Figs. 2c and 4c. As the sheet distortion increases, the corrugation increases as well and with it the force needed to slip over the barriers in the $x$ direction. This leads to the characteristic strengthening of the stick–slip motion seen in Figs. 2a and 4a. For sufficiently large sheet distortions $q$, the distortion energy $V_{\text{sheet}}(q)$ becomes so large that the rest of the minima in the trench disappear. Note that without the fourth-power contribution, however, this is not guaranteed. The tip will move along the trench, until it reaches the steady-state sheet distortion. After this, it can no longer move out further and instead starts slipping over the edges between the trenches and moving in the $x$ direction with the sheet distortion approximately constant. This explains also why the transition between the

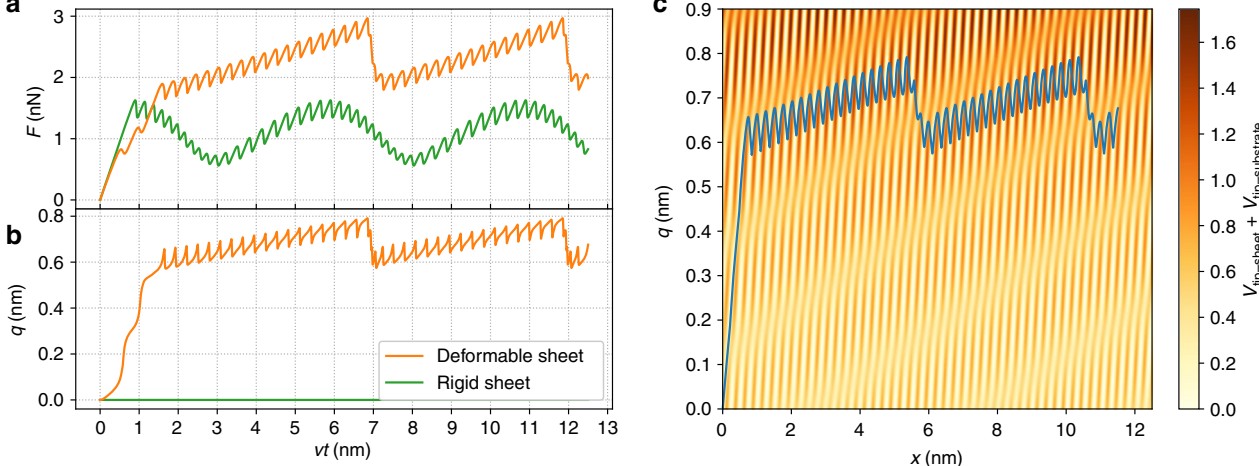

**Fig. 4 Strengthening for an incommensurate case.** Friction strengthening in a system with incommensurate lattice parameters but otherwise similar to Fig. 2, with (orange) and without (green) sheet deformation. The weakly incommensurate geometry of $b/a = 1.05$ corresponds to a moiré pattern with long-range periodicity of 20 nm. The steady-state lateral force shows the additional effect of the substrate, especially the long period for the moiré pattern. Otherwise the behaviour of the system is very similar to Fig. 2. However, the profile of the long-range periodicity is not sinusoidal, as would have been expected from a rigid sheet.

strengthening and steady-state regimes is abrupt, as has been observed in experiments[4].

In order to ensure the fidelity of this model under realistic conditions, we have also investigated the effect of thermal fluctuations and disordered substrates. In both cases, on average equivalent friction dynamics was obtained with some variations in stick–slip periods. In the case of thermal noise, this is due to thermally activated slips, which means that the tip may slip earlier due to noise, leading to lower friction (thermolubricity)[19]. In the case of a disordered, nonperiodic substrate, slips are also naturally not periodic.

**Role of the substrate**. We can now also investigate the role of the substrate. To this end, we remove the tip–substrate term in Eq. (1), as well as set $\nu_2$ to zero, since it is mainly the result of substrate adhesion. Figure 6a shows force traces for a system without a substrate. If we tune the parameters somewhat (details in the "Methods" section), we can recover force traces that look similar to those before. Nevertheless, the behaviour is still qualitatively different, as there is a dependence on velocity. Without the substrate, the strengthening is clearly not robust. In AFM experiments, the velocities are quite low, and without a substrate we would not expect the initial strengthening to appear, but the friction would still depend on the number of layers, as has indeed been found[5,12].

This behaviour can again be understood from the energy landscape. As can be seen from Fig. 6b, the slanted trenches are still present, but they no longer contain a sequence of energy minima. As a result, the $xq$-trajectory will now be governed by dynamics and not just the topology of the energy landscape. The tip simply immediately moves out to the maximum $q$ as fast as its inertia will allow. If the sliding velocity or inertia is low and the damping parameter is not high, the first slip will occur only when the sheet distortion has already approximately reached its steady-state value. For high sliding speeds, or large inertia $m_q$, the dynamics come into play and slips happen as $q$ is still increasing. This produces a force trace with somewhat similar looking strengthening. However, the slope of the strengthening is dependent on the velocity. This behaviour is not very robust, as it appears in a more narrow range of parameter values than the strengthening in the quasi-static case with a substrate.

**Analytical estimation of the friction**. We can estimate the steady-state friction analytically. To do this, we will neglect the substrate–tip term, since this is generally much weaker than the tip–sheet term and thus its impact on the steady-state sheet distortion and on the friction is small. Consider the moment just before a slip in the steady state. At this point, all forces are in

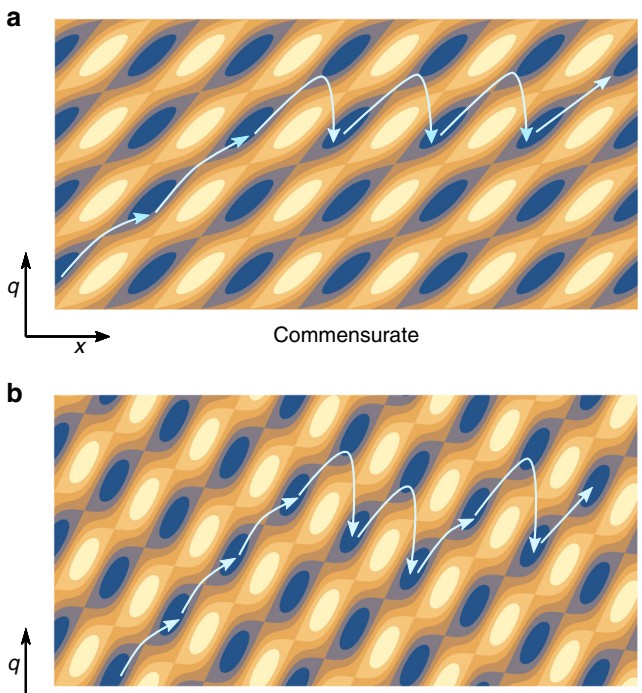

**Fig. 5 A cartoon explaining the strengthening behaviour.** The potential energy landscape and trajectory of the tip in the $xq$-plane for **a** commensurate and **b** incommensurate lattice parameters of the sheet and substrate. Blue indicates low values of the potential energy and yellow high values. We can understand the strengthening from the topology of this landscape.

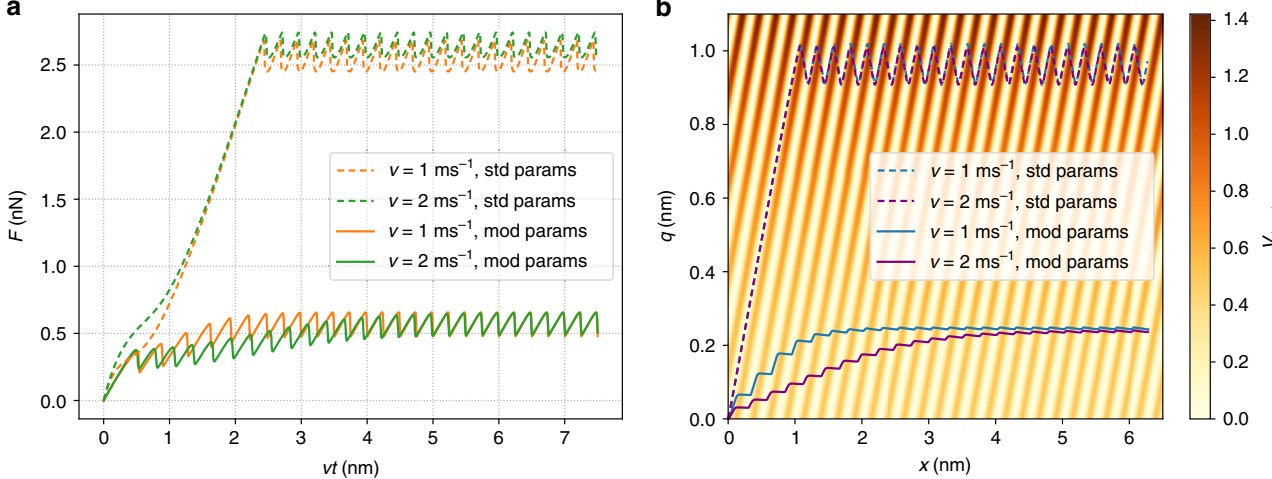

**Fig. 6 The case with no substrate.** Force traces (**a**) and $xq$-trajectories (**b**) for a system without a substrate. The frictional strengthening disappears for our chosen parameters (indicated by the abbrevaition "std params") but can be recovered by tuning the parameters (indicated by "mod params"). Details of the parameters are given in the "Methods" section. The strengthening without the substrate is dynamic rather than quasi-static in nature, such that it depends on the velocity as well as the inertia $m_q$.

balance, the sheet distortion is at its maximum value $q_{max}$, and the tip is almost at the inflection point of the potential in the $x$ direction. The forces in the $q$ direction at this point are

$$\left.\frac{\partial U}{\partial q}(x,q,t)\right|_{q=q_{max}} = 0 \qquad (5)$$

$$= 2\nu_2 q_{max} + 4\nu_4 q_{max}^3 + 2q_{max}\kappa_1\left[1 - \cos\left(\frac{2\pi}{a}(x - q_{max})\right)\right]$$
$$- \frac{2\pi}{a}(V_1 + \kappa_1 q_{max}^2)\sin\left(\frac{2\pi}{a}(x - q_{max})\right). \qquad (6)$$

To find the inflection point, we apply the condition $\partial^2 U/\partial q^2 = 0$, which is met when $\cos[2\pi(x - q)/a] = 0$ and $\sin[2\pi(x - q)/a] = 1$. Substituting this into Eq. (6), we find

$$2\nu_2 q_{max} + 4\nu_4 q_{max}^3 + 2\kappa_1 q_{max} - \frac{2\pi}{a}(V_1 + \kappa_1 q_{max}^2) = 0. \qquad (7)$$

This polynomial has only one real root, from which we can then obtain the force in the slip point through $F_{lat}^{max} = \frac{2\pi}{a}(V_1 + \kappa_1 q_{max}^2)$.

This gives a reasonable approximation of the maximum lateral force, which in turn gives an approximation of the friction. Figure 3 shows the steady-state force as a function of the parameter $\nu_4$ for both the analytical expression in Eq. (7) and numerical simulations for the incommensurate case with $a/b = \frac{1}{2} + \frac{1}{2}\sqrt{5}$. For a commensurate case, this would look very similar. Even though the substrate plays an important role in the strengthening, due to the weak contribution to the potential energy, it does not strongly affect $q_{max}$ and the steady-state friction.

## Discussion

The model we have introduced here captures the crucial part of the dynamics that gives rise to frictional strengthening and layer-number dependence of friction. It provides fundamental insight into the friction on sheets of layered materials and also into a range of results obtained over the past decade from experiments as well as detailed atomistic simulations. We find that the dynamics of the strengthening and layer-number dependence of friction is very universal. Our model furthermore quantitatively reproduces the typical behaviours that have been observed many times since the initial experiments[3,4].

The role of the extra degree of freedom can be played not only, for example, by out-of-plane pucking, smaller out-of-plane distortions, and distortion in the $xq$-plane[20], but also by more subtle issues, such as the quality of the contact described in ref. [10]. This explains the seemingly contradictory results from different experiments and simulations[5,6,8,10,11]. These different works have probed different mechanisms producing similar dynamics and consequently showed similar friction behaviour, even though at a more detailed level other degrees of freedom were at play. Moreover, there may be non-trivial relations between in-plane and out-of-plane distortions[21,22] that could contribute simultaneously.

From our model, it also becomes clear that the substrate plays an important role in the strengthening but less so in the layer dependence. The structure of the energy landscape is qualitatively changed by the weak interaction between the substrate and the tip through the sheet. This gives rise to extra minima in the energy landscape, which lead to the step-wise changes in the sticking conditions and lateral forces in the beginning.

When the substrate is removed, the strengthening behaves qualitatively very differently. It is controlled by the inertia of the distortion and is also more sensitive to parameters. Moreover, a high load on a suspended layer would lead to an increase in $\nu_2$,

and this in turn decreases $q_{max}$ (see Eq. (7)). This explains the unusual behaviour observed in both experiments[12] and atomistic simulations[13] of lower friction at high loads for suspended sheets. We are not aware of any experiments that have investigated the strengthening of friction on suspended sheets, only the layer dependence[5,12]. Based on our model, it is not obvious that it would be detectable under experimental conditions. However, if it is we expect it will show a dependence on the sliding velocity that is not present in systems with a substrate.

Another experiment we can explain now using our model is that of ref. [5] where no strengthening was observed on strong substrates. These substrates would have increased $\nu_2$ to such an extent that minima with nonzero distortion would no longer have been accessible. Yet another example of a perplexing experimental result that can be understood is given in ref. [7]. In that work, a velocity dependence is found of the layer-dependent friction, but the slope of the strengthening does not change. This is consistent with our model combined with thermal noise and is simply the effect of thermolubricity[19,23].

Furthermore, the insight we gain from this model opens up a lot of new avenues for exploring friction on thin sheets of layered materials. The role of the substrate lattice in frictional strengthening and layer-number dependence has not been investigated much experimentally, but there is currently a large interest in thin sheets on various substrates in the context of moiré patterns[18,24–26]. These have been suggested to allow for interesting tuning of friction through interaction parameters between the substrate and sheet (see, for example, ref. [27]). Using the framework of our model, we can now understand the interplay between the moiré pattern, frictional strengthening, and distortion of the sheet, which leads to a very rich phenomenology.

Our model is also a crucial step towards understanding extreme distortions and tearing of atomically thin sheets under high loads. In real conditions, such extreme distortions are common and lead to breaking of chemical bonds, tearing, wear, and loss of low-friction conditions. This distortion strength could be included in the future through a maximum allowed $q$. Wear is almost invariably more complex than friction itself, and development of new technology is usually done by trial and error. It would thus be worthwhile to further study wear using this model and include additional subtleties based on what is known from atomistic simulations in refs. [28,29]. Nevertheless, this is a first step and raises the possibility of better understanding of wear and faster, understanding-based, development of practical applications of graphene in low-friction technologies.

## Methods

**Equations of motion**. The time evolution of a system with the potential energy in Eq. (1) is governed by the equations of motion

$$\begin{cases} m_x\ddot{x} = -\frac{dU(x,q)}{dx} - m_x\eta_x\dot{x}, \\ m_q\ddot{q} = -\frac{dU(x,q)}{dq} - m_q\eta_q\dot{q}, \end{cases} \qquad (8)$$

where $m_x$ and $m_q$ are the $x$ and sheet distortion inertia, respectively, and $\eta_x$ and $\eta_q$ are the damping coefficients.

Many readily available implementations of the various algorithms for solving differential equations numerically can be used, since the equations of motion are not complicated. We obtain numerical solutions to the equations of motion using the *Mathematica* differential equation solver NDSolve[30]. The lateral force can then readily be calculated by $F_{lat}(t) = k(vt - x(t))$ and the friction is given by the time average of the lateral force.

**Parameters**. In the spirit of reproducibility, we have chosen parameter values estimated from the system used by Li et al. in their detailed atomistic simulations of friction on graphene sheets[10]. Unless otherwise stated, the following parameter values are used. The inertias $m_x = 501.40\ m_{carbon}$, $m_q = 179.07\ m_{carbon}$ are chosen to be consistent with the AFM scale. The damping parameters $\eta_x = 18.75\ ps^{-1}$ and $\eta_q = 42.86\ ps^{-1}$ are typical values for the atomic level. For the tip–support

interaction, $v = 1.0\,\mathrm{ms}^{-1}$, $k = 2.0\,\mathrm{Nm}^{-1}$, and $a = 2.5$ A, i.e. the graphene lattice period in the zig-zag direction.

Unless otherwise stated, we use $a/b = 1$, which corresponds to a commensurate surface. When we use an incommensurate ratio of lattice parameters, we use $a/b = (1 + \sqrt{5})/2$ (the golden ratio). The corrugation parameter $V_1 = 0.25\,\mathrm{eV}$ is a good estimate of the corrugation of a small tip on graphene, while $V_2 = 0.125\,\mathrm{eV}$ is simply half of that to account for the masking by the sheet in between the tip and substrate. The corrugation coefficients $\kappa_1 = 0.375\,\mathrm{eV\,nm}^{-2}$ and $\kappa_2 = 0.1875\,\mathrm{eV\,nm}^{-2}$ are chosen with similar considerations to be the right order of magnitude for corrugation effects. The quadratic leading order in the distortion energy is related to the sheet binding to the substrate and is chosen to represent the order of magnitude of the adhesion of an AFM tip-sized sheet of graphene on the substrate, $\nu_2 = 2.39\,\mathrm{eV\,nm}^{-2}$. The fourth-order term in the distortion has the order of magnitude given by distorting covalent bonds inside a single layer of graphene, $\nu_4 = 3.64\,\mathrm{eV\,nm}^{-4}$.

When we investigate the system in the absence of a substrate, we have tuned the parameters to recover the strengthening behaviour. In that case, we have used $V_1 = 0.08\,\mathrm{eV}$, $V_2 = 0.0\,\mathrm{eV}$, $\kappa_1 = 0.025\,\mathrm{eV\,nm}^{-2}$, and $\nu_4 = 0.021\,\mathrm{eV\,nm}^{-4}$.

## Data availability
Data are available from the authors upon request.

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

## Acknowledgements
D.A. acknowledges Anders Lundkvist for insightful discussions. This work has been supported financially by the Swedish Research Council (Vetenskapsrådet) grant number 2015-04962 and by the COST action MP1303.

## Author contributions
A.S.d.W. conceived of the study. D.A. performed the simulations. The authors analysed the results and wrote the manuscript together.

## Competing interests
The authors declare no competing interests.
