## [Peer Review File · Nature Communications]

Reviewers' comments:

Reviewer #1 (Remarks to the Author):

The paper presents a simple model for friction on atomically thin sheets, which is currently a hot research topic. Experiments have for instance shown a decreasing friction coefficient with increasing number of layers. One can conduct MD simulations to uncover the mechanistic origins. Here the authors upgrade/modify a much simpler Prandtl-Tomlinson (PT) model to explain observables, and this with remarkable success. The idea is to add an extra degree of freedom to “isolate and understand the dynamics of strengthening and layer-number dependence of friction”.

I was surprised that no one attempted this simple modification before, especially because PT models have enjoyed quite some success in the literature. It is a very popular and classical approach. It seems to me that the authors found a clever and original application of PT model.

Of course one could argue that with such simple models you get what you put in. In the words of the paper: “if we assume that each layer is distorted in roughly the same way, the energy penalty should grow roughly linearly with the number of layers n , and thus v_4 is proportional to n ”. This choice for the v_4 parameter forces the dependence on the number of layers. Then, we wonder here if second order effects might be missed. These would be present in more sophisticated models, including MD simulations. Similarly I was not convinced by the sentence “if we tune the parameters somewhat, we can recover...”. A better justification is needed.

Also, a limit of the simple approach taken here is that it certainly cannot account for what is precisely the nature of the extra degree of freedom, and gives little room for design. Is it out-of-plane pucking, smaller out-of-plane distortions, distortion in the xq -plane, or contact quality? Or as the authors put it “there may be non-trivial relations between in-plane and out-of-plane distortions that could contribute simultaneously”.

Nevertheless I find beauty in the model's simplicity. I believe it reveals key ingredients (and global dynamics) that have been overlooked in past discussions. I therefore recommend accepting the paper for publication. Overall the paper is interesting, clear and easy to follow. It should generate interesting discussions.

I would urge the authors to go through a careful proof reading as

I could find a number of typos.

Examples:

“grahpene” in fig 3

Purple circle symbol in fig. 3 not explained

Repeated “the” p.3

“substate” p. 10

“trail-and-error”

JF Molinari

Reviewer #2 (Remarks to the Author):

Anderson and de Wijn suggest a simple model for the friction of atomically thin layered materials on substrates. This model is based on the Prandtl-Tomlinson ansatz extended by an addition degree of freedom q for the thin sheet. If the 4th order penalty for increasing q is related to the sheet thickness, the authors' model predicts a decrease of friction with sheet thickness that has been found in several experiments and simulations. Furthermore, their model reproduces frictional strengthening of the system also in agreement with publications in literature. The former effect occurs for different experimental systems and different mechanisms are reported that underlie the smaller friction on thicker sheets (such as puckering or change of contact quality). Anderson and de Wijn provides fundamental insights into the observed frictional responses independent of the microscopic processes. They do this quite elegantly by discussing the two-dimensional potential energy landscape in the q,x plane.

I found the simple model described in this manuscript intriguing and think that this work is well suited for Nature Communications. However, there are a couple of points I would like to be addressed before I can recommend publication.

Although the manuscript contains all information to understand the main flow of arguments, there are some claims that are not supported by data. The authors should add Supporting Material for the following cases:

A) More details about the exact choice of model parameters. How are the parameters that have not been extracted from literature been obtained? How sensitive are the frictional responses to reasonable variations of these parameters? Why were parameters changed for sheets without a substrate? Was it just to see a desired behavior (as indicated in the manuscript) or was there also a physical reasoning.

B) The authors discuss thermolubric effects without showing data. Please describe in detail.

C) The description of the analytical model is quite short. Please provide an extended derivation.

D) How would Fig. 3 change for commensurate contacts?

E) Another point is the dependence of the tip-substrate interaction on sheet thickness. How is it included in Eq. (4)?

I see the usefulness of this model for predicting friction of thin layered materials. However, the relation to wear the authors try to make in the outlook is not obvious to me. There are important experimental and molecular dynamics works that show how complicated wear of these materials can be (e.g. Moseler&Bennewitz or work by T.B.Ma). How does the current model related to breaking of chemical bonds in layered materials or rebonding to the substrate?

Reviewer #3 (Remarks to the Author):

The manuscript "Understanding the friction of atomically thin layered materials" by Anderson and de Wijn address the question why the lubrication of graphite is sensitive to the number of layers. I feel that this question is already well understood, but perhaps not yet translated very well for scientists having the intellectual horizon of the Prandtl-Tomlinson (PT) community. I am sure that this very large community will celebrate the submitted contribution and that the paper has the potential to collect many citations, which is all Nature Communication actually cares about. In that sense, and also because I feel that the conclusions drawn in the paper are qualitatively correct, I can certainly recommend publication.

At the same time, I must say that I am a bit disappointed by the work. First, it does not credit literature discussing that in-plane elasticity increases friction, while out-of-plane compliance reduces it. Effectively, it is contained in a classical work by Shinjo and Hirano, where they write in the abstract (!) of 10.1016/0039-6028(93)91022-H:

"It is emphasized that a high dimensionality in the friction system is a key to understanding the physics of superlubricity." Later works by varying authors elucidated this aspect and demonstrated

perhaps more clearly the competing roles of in-plane and out-of-plane compliance, as, e.g., in 10.1103/PhysRevLett.86.1295.

In addition, I am not sure if the coarse-graining, which the authors effectively do, is really the leading-order coarse-graining to be done. Starting point should have been a general Steele potential, i.e., $V(z) + V_1(z) \sum_{\mathbf{g}} \cos(\mathbf{g} \cdot \mathbf{r})$, where \mathbf{g} are the reciprocal lattice vectors of the substrate, and $V(z)$ and $V_1(z)$ coefficients that depend on size and orientation of the graphite flakes, and an elasticity that can differ in in-plane and out-of-plane directions (de Wijn has actually very good papers that implicitly address the question how $V(z)$ and $V_1(z)$ depend on size and orientation of flakes). I believe, though I am not 100% certain, that the leading-order corrections to the PT model coming from this quite systematic approach differ from the expressions proposed by the authors. However, I am certain that the trends (not the actual numbers) will be similar. This scepticism includes the question if thickness really affect only the q^4 term and not the q^2 term.

In conclusion, I feel that the paper ranked against many of the highly-cited PT works is very good. However, the discussion should reflect the state of the art outside of the PT community. And the model can certainly be better motivated from a more systematic coarse-graining aspect.

Reply to referee comments

David Andersson and Astrid S. de Wijn

November 28, 2019

We believe we have addressed the issues raised by Reviewers #1 and #2. With regard to the comments of Reviewer #3, there seems to be a misunderstanding about what system we are studying. We discuss all of this in more detail below, and have made clarifications to the manuscript.

We have improved the figures to comply with nature's manuscript style policies.

> Reviewer #1 (Remarks to the Author):

>

> The paper presents a simple model for friction on atomically thin sheets,
> which is currently a hot research topic. Experiments have for instance
> shown a decreasing friction coefficient with increasing number of layers.
> One can conduct MD simulations to uncover the mechanistic origins. Here
> the authors upgrade/modify a much simpler Prandtl-Tomlinson (PT) model to
> explain observables, and this with remarkable success. The idea is to add
> an extra degree of freedom to "isolate and understand the dynamics of
> strengthening and layer-number dependence of friction".

>

> I was surprised that no one attempted this simple modification before,
> especially because PT models have enjoyed quite some success in the
> literature. It is a very popular and classical approach. It seems to me
> that the authors found a clever and original application of PT model.

We are happy that the referee seems to appreciate our model. We were also surprised by the simplicity.

> Of course one could argue that with such simple models you get what you
> put in. In the words of the paper: "if we assume that each layer is
> distorted in roughly the same way, the energy penalty should grow roughly
> linearly with the number of layers n , and thus ϵ is proportional to n ".
> This choice for the ν_4 parameter forces the dependence on the number of
> layers. Then, we wonder here if second order effects might be missed.

The parameter ϵ is already the next to leading order. Without it the model behaves qualitatively differently, as we note in the manuscript.

We actually study the behaviour as a function of ν_4 (see equation 6), not directly as a function of the number of layers, so higher-order terms in n would only appear in a local rescaling of the

x axis in figure 3. Since the curve is nearly horizontal for thick sheets anyway, this will not cause any qualitative difference. We have added a comment to this effect to the manuscript.

- > These would be present in more sophisticated models, including MD
- > simulations. Similarly I was not convinced by the sentence "if we tune the
- > parameters somewhat, we can recover...". A better justification is needed.

Our point with this statement is in fact that while with the substrate the strengthening is very general, without the substrate it is not. The need for tuning the parameters in the substrate-less case demonstrates that the substrate is crucial for the strengthening behaviour and without it the strengthening is not robust. We therefore do not expect that experiments on free-hanging sheets would show the same strengthening in the beginning of sliding, even though the friction on thin sheets would still be higher. We have added a comment to the manuscript to emphasise this.

- > Also, a limit of the simple approach taken here is that it certainly
- > cannot account for what is precisely the nature of the extra degree of
- > freedom, and gives little room for design. Is it out-of-plane pucking,
- > smaller out-of-plane distortions, distortion in the xq -plane, or contact
- > quality? Or as the authors put it "there may be non-trivial relations
- > between in-plane and out-of-plane distortions that could contribute
- > simultaneously".

An important point that we are trying to make is that it can be any of these things. Previous work has debated heavily which of these different degrees of freedom were responsible. Different authors have shown quite convincingly that different degrees of freedom were at play in their different systems. This has led to seemingly contradictory conclusions about which degrees of freedom were the cause of the strengthening. We show in this work that there is in fact no single answer. These different mechanisms are not mutually exclusive. Different degrees of freedom are playing the same role of q in different atomistic simulations and experiments. We already noted this in the last sentence of the introduction and the second paragraph of the discussion section. We have added some extra emphasis on this point in the manuscript.

- > Nevertheless I find beauty in the model's simplicity. I believe it reveals
- > key ingredients (and global dynamics) that have been overlooked in past
- > discussions. I therefore recommend accepting the paper for publication.
- > Overall the paper is interesting, clear and easy to follow. It should
- > generate interesting discussions.
- >
- > I would urge the authors to go through a careful proof reading as
- >
- > I could find a number of typos.
- > Examples:
- > "grahpene" in fig 3
- > Purple circle symbol in fig. 3 not explained
- > Repeated "the" p.3
- > "substate" p. 10
- > "trail-and-error"

We have proofread the entire manuscript again, and fixed all typos we could find.

The purple circle in figure 3 was not real. It was two purple crosses at 45 degree angle on top of each other that ended up looking like a purple circle. This is difficult to prevent due to the rescaling on the y axis. We have added a comment to the caption to prevent confusion.

> JF Molinari

>

> Reviewer #2 (Remarks to the Author):

>

> Anderson and de Wijn suggest a simple model for the friction of atomically
> thin layered materials on substrates. This model is based on the
> Prandtl-Tomlinson ansatz extended by an addition degree of freedom q for
> the thin sheet. If the 4th order penalty for increasing q is related to
> the sheet thickness, the authors' model predicts a decrease of friction
> with sheet thickness that has been found in several experiments and
> simulations. Furthermore, their model reproduces frictional strengthening
> of the system also in agreement with publications in literature. The
> former effect occurs for different experimental systems and different
> mechanisms are reported that underlie the smaller friction on thicker
> sheets (such as puckering or change of contact quality). Anderson and de
> Wijn provides fundamental insights into the observed frictional responses
> independent of the microscopic processes. They do this quite elegantly by
> discussing the two-dimensional potential energy landscape in the
> q, x plane.

> I found the simple model described in this manuscript intriguing and think
> that this work is well suited for Nature Communications.

We are happy that this referee appreciates our model.

> However, there

> are a couple of points I would like to be addressed before I can recommend
> publication.

> Although the manuscript contains all information to understand the main
> flow of arguments, there are some claims that are not supported by data.

> The authors should add Supporting Material for the following cases:

> A) More details about the exact choice of model parameters. How are the
> parameters that have not been extracted from literature been obtained? How
> sensitive are the frictional responses to reasonable variations of these
> parameters? Why were parameters changed for sheets without a substrate?
> Was it just to see a desired behavior (as indicated in the manuscript) or
> was there also a physical reasoning.

As stated in the methods section, most parameters were directly extracted from Li et al's atomistic simulations. This includes all parameters related to the energy and length scales. Others (the damping parameters) are on the order of magnitude that is generally used in the literature. The only parameters we have chosen ourselves are the masses to be consistent with AFM experiments. Due to the nature of the PT model, however, it is only the energy and length scales that

control the stick-slip behaviour, and in this model the strengthening. While this was already written in the text, so we have added some extra comments to further clarify it.

Regarding the substrate-less case, please see our reply to the first referee above.

> B) The authors discuss thermolubric effects without showing data. Please
> describe in detail.

Thermally activated slips and the thermolubric effect are actually rather subtle, but also well-studied for the simple PT model and in experiments. We have added a comment to elaborate and a citations to some of the most fundamental literature on this topic.

> C) The description of the analytical model is quite short. Please provide
> an extended derivation.

For clarity, we have added one intermediate step. Besides what is now shown, this derivation is just some simple algebra.

> D) How would Fig. 3 change for commensurate contacts?

It would not change. The periodicity of the substrate plays an important role in the strengthening, but not in the steady-state friction. We have added a comment about this to the manuscript.

> E) Another point is the dependence of the tip-substrate interaction on
> sheet thickness. How is it included in Eq. (4)?

It would be included through V_2 and κ_2 . However, the effect of the substrate on the steady-state friction is negligible regardless of the layer thickness. This was already noted in the derivation in the manuscript.

> I see the usefulness of this model for predicting friction of thin layered
> materials. However, the relation to wear the authors try to make in the
> outlook is not obvious to me. There are important experimental and
> molecular dynamics works that show how complicated wear of these materials
> can be (e.g. Moseler&Bennewitz or work by T.B.Ma). How does the current
> model related to breaking of chemical bonds in layered materials or
> rebonding to the substrate?

Our model is only a first step in this direction. Wear is almost invariably more complex than friction itself. The breaking of chemical bonds could be considered in the context of this model as a maximum allowed distortion of the sheet. If q exceeds this value, the sheet is torn. Clearly, additional subtleties of wear would have to be included. We have added a few extra words and qualifiers to the final paragraph to clarify this and included the references that the referee suggests.

> Reviewer #3 (Remarks to the Author):
 >
 > The manuscript "Understanding the friction of atomically thin layered
 > materials" by Anderson and de Wijn address the question why the
 > lubrication of graphite is sensitive to the number of layers. I feel that
 > this question is already well understood, but perhaps not yet translated
 > very well for scientists having the intellectual horizon of the
 > Prandtl-Tomlinson (PT) community. I am sure that this very large community
 > will celebrate the submitted contribution and that the paper has the
 > potential to collect many citations, which is all Nature Communication
 > actually cares about. In that sense, and also because I feel that the
 > conclusions drawn in the paper are qualitatively correct, I can certainly
 > recommend publication.

We are happy that the referee recommends publication, but would like to point out that also experimental scientists and people performing atomistic simulations are actively debating the nature of this effect.

> At the same time, I must say that I am a bit disappointed by the work.
 > First, it does not credit literature discussing that in-plane elasticity
 > increases friction, while out-of-plane compliance reduces it. Effectively,
 > it is contained in a classical work by Shinjo and Hirano, where they write
 > in the abstract (!) of 10.1016/0039-6028(93)91022-H:
 > "It is emphasized that a high dimensionality in the friction system is a
 > key to understanding the physics of superlubricity." Later works by
 > varying authors elucidated this aspect and demonstrated perhaps more
 > clearly the competing roles of in-plane and out-of-plane compliance, as,
 > e.g., in 10.1103/PhysRevLett.86.1295.

Both works that the referee refers to deal with quite different systems where two extended objects slide relative to each other, f.e. a sheet sliding on a substrate. In our current work we deal with a tip sliding on a sheet that lies stationary on the substrate. In our case, the two extended objects do not slide relative to each other. This means that the structure of the sheet and substrate relate to each other in a completely different way. We believe the referee may be confused about this distinction due to a lack of clarity in figure 1. We have added some clarification to the caption.

For extended sliding objects like the referee seems to have in mind, one would often consider models more in the direction of the Frenkel-Kontorova model and similar constructions, rather than the Prandtl-Tomlinson model. The two papers that the referee refers to are certainly extremely important works in the literature on friction of extended sliding objects, but they cannot be translated to the problem we consider here.

> In addition, I am not sure if the coarse-graining, which the authors
 > effectively do, is really the leading-order coarse-graining to be done.
 > Starting point should have been a general Steele potential, i.e., $V(z) +$
 > $V_1(z) \sum_g \cos(g \cdot r)$, where g are the reciprocal lattice vectors of the
 > substrate, and $V(z)$ and $V_1(z)$ coefficients that depend on size and
 > orientation of the graphite flakes, and an elasticity that can differ in

> in-plane and out-of-plane directions (de Wijn has actually very good
> papers that implicitly address the question how $V(z)$ and $V_1(z)$ depend on
> size and orientation of flakes). I believe, though I am not 100% certain,
> that the leading-order corrections to the PT model coming from this quite
> systematic approach differ from the expressions proposed by the authors.
> However, I am certain that the trends (not the actual numbers) will be
> similar. This scepticism includes the question if thickness really affect
> only the q^4 term and not the q^2 term.

The work of de Wijn that the referee refers to also deals with extended sliding objects and is thus of a different nature as well. Like the works by Shinjo and Hirano and Müser et al, it deals with an extended object (a graphene flake) sliding on a substrate, not a tip sliding on a stack of extended objects. Thus the coarse-graining approach from that previous work, and the coarse-graining approach that the referee suggests, are both not applicable here. Any distortion of the type that the referee seems to be suggesting would be included already in the potential energy for $q = 0$.

For the problem we consider in this work, what we have done is actually a systematic coarse-graining by using an expansion of the potential energy up to next to leading order in the distortion q . We have added additional comments about this to the text.

We do not expect that the $\nu_2 q^2$ term depends strongly on the number of layers, since under realistic conditions, it is dominated by the adhesion between the sheet and substrate, not the bending of the sheet. Moreover, the steady-state distortion and friction are anyway controlled by the q^4 term, not the q^2 term. We have added additional comments to the text.

> In conclusion, I feel that the paper ranked against many of the
> highly-cited PT works is very good. However, the discussion should reflect
> the state of the art outside of the PT community. And the model can
> certainly be better motivated from a more systematic coarse-graining
> aspect.

Our coarse-graining approach is systematic, but this was apparently not clearly explained. We have added several comments to clarify this.